# Federated Reconstruction:
# Partially Local Federated Learning

**Karan Singhal**
Google Research
karansinghal@google.com

**Hakim Sidahmed**
Google Research
hsidahmed@google.com

**Zachary Garrett**
Google Research
zachgarrett@google.com

**Shanshan Wu**
Google Research
shanshanw@google.com

**Keith Rush**
Google Research
krush@google.com

**Sushant Prakash**
Google Research
sush@google.com

## Abstract

Personalization methods in federated learning aim to balance the benefits of federated and local training for data availability, communication cost, and robustness to client heterogeneity. Approaches that require clients to communicate all model parameters can be undesirable due to privacy and communication constraints. Other approaches require always-available or stateful clients, impractical in large-scale cross-device settings. We introduce Federated Reconstruction, the first model-agnostic framework for partially local federated learning suitable for training and inference at scale. We motivate the framework via a connection to model-agnostic meta learning, empirically demonstrate its performance over existing approaches for collaborative filtering and next word prediction, and release an open-source library for evaluating approaches in this setting. We also describe the successful deployment of this approach at scale for federated collaborative filtering in a mobile keyboard application.

## 1 Introduction

Federated learning is a machine learning setting in which distributed clients solve a learning objective on sensitive data via communication with a coordinating server [44]. Typically, clients collaborate to train a single global model under an objective that combines heterogeneous local client objectives. For example, clients may collaborate to train a next word prediction model for a mobile keyboard application without sharing sensitive typing data with other clients or a centralized server [28]. This paradigm has been scaled to production and deployed in cross-device settings [3, 28, 56] and cross-silo settings [11, 13].

However, training a fully global federated model may not always be ideal due to heterogeneity in clients' data distributions. Yu et al. [58] show that global models can perform worse than purely local (non-federated) models for many clients (*e.g.,* those with many training examples). Moreover, in some settings privacy constraints completely prohibit fully global federated training. For instance, for models with user-specific embeddings, such as matrix factorization models for collaborative filtering [37], naively training a global federated model involves sending updates to user embeddings on the server, directly revealing potentially sensitive individual preferences [21, 47].

To address this, we explore partially local federated learning. In this setting, models are partitioned into global $g$ and local parameters $l$ such that local parameters never leave client devices. This enables training on sensitive user-specific parameters as in the collaborative filtering setting, and we show it can also improve robustness to client data heterogeneity and communication cost for other settings,

35th Conference on Neural Information Processing Systems (NeurIPS 2021).

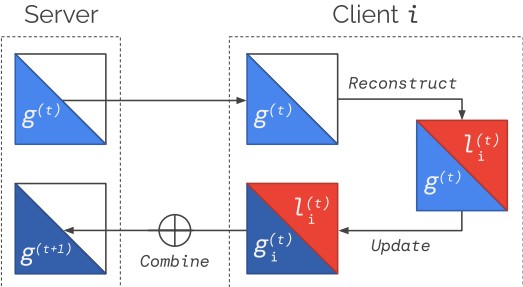

Figure 1: Schematic of Federated Reconstruction. Model variables are partitioned into global and local variables. For every round $t$, each participating client $i$ is sent the current global variables, uses them to reconstruct its own local variables, and then updates its copy of the global variables. The server aggregates updates to only the global variables across clients.

since we are effectively interpolating between local and federated training. Previous works have looked at similar settings [4, 41]. Importantly, these approaches cannot realistically be applied at scale in cross-device settings because they assume clients are stateful or always-available: in practice, clients are sampled from an enormous population with unreliable availability, so approaches that rely on repeated sampling of the same stateful clients are impractical (Kairouz et al. [34] [Table 1]). Other work has demonstrated that stateful federated algorithms in partial participation regimes can perform worse than stateless algorithms due to the state becoming "stale" [48]. Previous methods also do not enable inference on new clients unseen during training, preventing real-world deployment.

These limitations motivate a new method for partially local federated learning, balancing the benefits of federated aggregation and local training. This approach should be:

1. *Model-agnostic:* works with any model.
2. *Scalable:* compatible with large-scale cross-device training with partial participation.
3. *Practical for inference:* new clients can perform inference.
4. *Fast:* clients can quickly adapt local parameters to their personal data.

In this work, we propose combining federated training of global parameters with *reconstruction* of local parameters (see Figure 1). We show that our method relaxes the statefulness requirement of previous work and enables fast personalization for unseen clients without additional communication, even for models without user-specific embeddings.

**Our contributions:** We make the following key contributions:

- Introduce a model-agnostic framework for training partially local and partially global models, satisfying the above criteria. We propose a practical algorithm instantiating this framework (FEDRECON).
- Justify the algorithm via a connection to model-agnostic meta learning (see Section 4.2), showing that FEDRECON naturally leads to fast reconstruction at test time (see Table 1).
- Demonstrate FEDRECON's empirical performance over existing approaches for applications in collaborative filtering and next word prediction, showing that our method outperforms standard centralized and federated training in performance on unseen clients (see Table 1), enables fast adaptation to clients' personal data (see Figure 3), and matches the performance of other federated personalization techniques with less communication (see Figure 2).
- Release an open-source library for evaluating algorithms across tasks in this setting.[1]
- Describe the successful deployment of this approach at scale for collaborative filtering in a real-world mobile keyboard application (see Section 7).

## 2 Related Work

Previous works have explored personalization of federated models via finetuning [52, 58], meta learning / bi-level optimization [10, 16, 18, 33], and model interpolation [14, 27, 43]. Some works aim to improve training convergence with heterogeneous client gradient updates [35, 39], while

---

[1] https://git.io/federated_reconstruction

others address client resource heterogeneity [15, 49]. All of these approaches require communicating all client parameters during training, which can be unreasonable due to privacy and communication constraints for some models (discussed further in Section 3), which motivates methods that aggregate only part of a model as in our work.

Arivazhagan et al. [4] and Liang et al. [41] aggregate part of a model, but these approaches do not meet the criteria from Section 1. Similar to other works proposing local parameters [22, 31, 40], both approaches require clients to maintain local models across rounds, which is problematic when sampling clients from large populations (criterion 2). Arivazhagan et al. [4] assumes that all clients are available for training at all times and do not propose a method for performing inference on new clients (criterion 3). Liang et al. [41] requires new inference clients to be able to ensemble the outputs of all other clients' local models to evaluate on new data, which is unrealistic in practice due to communication and privacy constraints (criterion 3). These constraints are crucial: with previous methods most clients do not have a practical way to perform inference. Previous methods were also proposed for specific model types (criterion 1): Arivazhagan et al. [4] explores personalization layers after shared base layers and Liang et al. [41] learns personal representations of local data. Finally, as we discuss in Section 4.2, our method optimizes a meta learning objective for training global parameters that lead to fast reconstruction (criterion 4).

**Federated Collaborative Filtering:** We evaluate our approach on collaborative filtering [37] in Section 5.1.1. Prior work has explored federated matrix factorization: Ammad-Ud-Din et al. [2] avoids sending the user matrix to the server by storing it locally, aggregating only the item matrix globally. Chai et al. [9] applies homomorphic encryption to aggregation of the item matrix. Flanagan et al. [20] studies federated collaborative filtering as a multi-view learning problem. Each approach requires clients to maintain state, unlike our method. Ammad-Ud-Din et al. [2] and Chai et al. [9] also do not address the problem of inference on unseen users.

**Federated Meta Learning:** Our approach is motivated by a connection to meta learning, described in Section 4.2. Other federated learning works have also established connections to meta learning: Jiang et al. [33] observed that training a global federated model that can be easily personalized via finetuning can be studied in the model-agnostic meta learning (MAML) framework [19], and FEDAVG is performing the distributed version of the REPTILE meta learning algorithm presented by Nichol et al. [46]. Chen et al. [10], Fallah et al. [18], and Lin et al. [42] apply the MAML algorithm and variants in federated settings. Khodak et al. [36] aims to improve upon these methods by learning client similarities adaptively. These methods do not address the partially local federated learning setting, where some parameters are not aggregated globally.

## 3 Partially Local Federated Learning

Typically, federated learning of a global model optimizes:

$$\min_{\mathbf{x}\in\mathbb{R}^d} F(\mathbf{x}) = \mathbb{E}_{i\sim\mathcal{P}}[f_i(\mathbf{x})] \tag{1}$$

where $f_i(\mathbf{x}) = \mathbb{E}_{\xi\in\mathcal{D}_i}[f_i(\mathbf{x}, \xi)]$ is the local objective for client $i$, $\mathbf{x}$ is the $d$-dimensional model parameter vector, $\mathcal{P}$ is the distribution of clients, and $\xi$ is a data sample drawn from client $i$'s data $\mathcal{D}_i$. In practical cross-device settings, $f_i(\mathbf{x})$ may be highly heterogeneous for different $i$, and the number of available clients may be large and constantly changing due to partial availability. Only a relatively small fraction of clients may be sampled for training.

To motivate partially local federated learning, we begin by considering models that can be partitioned into user-specific parameters and non-user-specific parameters. An example is matrix factorization in the collaborative filtering setting [30, 37]: in this scenario, a ratings matrix $R \in \mathbb{R}^{U \times I}$ representing user preferences is factorized into a user matrix $P \in \mathbb{R}^{U \times K}$ and an items matrix $Q \in \mathbb{R}^{I \times K}$ such that $R \approx PQ^\top$, where $U$ is the number of users and $I$ is the number of items. For each user $u$, this approach yields a $K$-dimensional user-specific embedding $P_u$.

To train this type of model in the federated setting, we cannot naively use the popular FEDAVG algorithm [44] or other (personalized) algorithms that involving aggregation of all model parameters. A simple application of global learning algorithms might require every client to be sent every other client's personal parameters, which is clearly unreasonable for both privacy and communication. A more sophisticated approach might be to have each client communicate only their own personal parameters with the server. In this case, the server still has access to individual user parameters,

---
**Algorithm 1** Federated Reconstruction Training
---

**Input:** set of global parameters $\mathcal{G}$, set of local parameters $\mathcal{L}$, dataset split function $S$, reconstruction algorithm $R$, client update algorithm $U$

**Server executes:**
    $g^{(0)} \leftarrow$ (initialize $\mathcal{G}$)
    **for** each round $t$ **do**
        $\mathcal{S}^{(t)} \leftarrow$ (randomly sample $m$ clients)
        **for** each client $i \in \mathcal{S}^{(t)}$ **in parallel do**
            $(\Delta_i^{(t)}, n_i) \leftarrow$ ClientUpdate$(i, g^{(t)})$
        **end for**
        $n = \sum_{i \in \mathcal{S}^{(t)}} n_i$
        $g^{(t+1)} \leftarrow g^{(t)} + \eta_s \sum_{i \in \mathcal{S}^{(t)}} \frac{n_i}{n} \Delta_i^{(t)}$
    **end for**

**ClientUpdate:**
    $(\mathcal{D}_{i,s}, \mathcal{D}_{i,q}) \leftarrow S(\mathcal{D}_i)$
    $l_i^{(t)} \leftarrow R(\mathcal{D}_{i,s}, \mathcal{L}, g^{(t)})$
    $g_i^{(t)} \leftarrow U(\mathcal{D}_{i,q}, l_i^{(t)}, g^{(t)})$
    $\Delta_i^{(t)} \leftarrow g_i^{(t)} - g^{(t)}$
    $n_i \leftarrow |\mathcal{D}_{i,q}|$
    return $\Delta_i^{(t)}, n_i$ to the server

---

which in this setting can be trivially used to recover sensitive user-item affinities, negating the privacy benefit of not centralizing the data (again unreasonable).

Thus a practical federated learning algorithm for this setting should be *partially local*: it should enable clients to train a subset of parameters entirely on-device. However, approaches that involve stateful clients storing their local parameters across rounds are undesirable in large-scale cross-device settings since clients are unlikely to be sampled repeatedly, causing state to be infrequently available and become stale, degrading performance (Reddi et al. [48] [Sec. 5.1]). Additionally, since only a fraction of clients participate in training, all other clients will be left without trained local parameters, preventing them from performing inference using the model. In a large population setting with hundreds of millions of clients as described in Section 7, this can mean 99%+ of clients do not have a complete model, preventing practical deployment. Thus an algorithm for this setting ideally should not depend on stateful clients and should provide a way to perform inference on unseen clients.

Though we have motivated partially local federated learning via a setting that contains privacy-sensitive user-specific parameters, we will later show that this paradigm can also improve robustness to heterogeneity in $f_i(\mathbf{x})$ and reduce communication cost, even for models without user-specific parameters. In this case, the partition between local and global parameters is determined by the use-case and communication limitations. As an example, in Section 5.1.2 we motivate a next word prediction use-case, where having a partially local model can be useful for handling diverse client inputs while reducing communication.

Achieving partially local federated learning in a practical cross-device setting with large, changing client distribution $\mathcal{P}$ and stateless clients is one of the key contributions of our work.

## 4 Federated Reconstruction

We now introduce the Federated Reconstruction framework. One of the key insights of our approach is that we can relax the requirement for clients to maintain local parameters across rounds by reconstructing local parameters whenever needed, running a reconstruction algorithm $R$ to recover them. Once a client is finished participating in a round, it can discard its reconstructed local parameters. An overview is presented in Figure 1.

Federated Reconstruction training is presented in Algorithm 1. Training proceeds as follows: for each round $t$, the server sends the current global parameters $g^{(t)}$ to each selected client. Selected clients split their local data $\mathcal{D}_i$ into a support set $\mathcal{D}_{i,s}$ and a query set $\mathcal{D}_{i,q}$. Each client uses its support set $\mathcal{D}_{i,s}$ and $g^{(t)}$ as inputs to reconstruction algorithm $R$ to produce its local parameters $l_i^{(t)}$. Then each client then uses its query set $\mathcal{D}_{i,q}$, its local parameters $l_i^{(t)}$, and the global parameters $g^{(t)}$ as inputs to update algorithm $U$ to produce updated global parameters $g_i^{(t)}$. Finally, the server aggregates updates to global parameters across clients. We describe key steps in further detail below.

**Dataset Split Step:** Clients apply a dataset split function $S$ to their datasets $\mathcal{D}_i$ to produce a support set $\mathcal{D}_{i,s}$ used for reconstruction and a query set $\mathcal{D}_{i,q}$ used for updating global parameters. Typically these sets are disjoint to maximize the meta-generalization ability of the model (see Section 4.2), but

in Appendix D we show that this assumption may be relaxed if clients don't have sufficient data to partition.

**Client Reconstruction Step:** Reconstruction of local parameters is performed by algorithm $R$. Though this algorithm can take other forms, in this work we instantiate $R$ as performing $k_r$ local gradient descent steps on initialized local parameters with the global parameters frozen, using the support set $\mathcal{D}_{i,s}$. We show in Section 4.2 this naturally optimizes a well-motivated meta learning objective. Interestingly, this approach is related to gradient-based alternating minimization, a historically successful method for training factored models [26, 32].

A potential concern with reconstruction is that this may lead to additional client computation cost compared to storing local parameters on clients. However, since clients are unlikely to be reached repeatedly by large-scale cross-device training, in practice this cost is similar to the cost of initializing these local parameters and training them with stateful clients. Additionally, reconstruction provides a natural way for new clients unseen during training to produce their own partially local models offline (see Section 4.1)–without this step, the vast majority of clients would not be able to use the model. Finally, in Section 4.2 we argue and in Section 5.2 we empirically demonstrate that with our approach *just one local gradient descent step* can yield successful reconstruction because global parameters are being trained for fast reconstruction of local parameters.

**Client Update Step:** Client updates of global parameters are performed by update algorithm $U$. In this work we instantiate $U$ as performing $k_u$ local gradient descent steps on the global parameters, using the query set $\mathcal{D}_{i,q}$.

**Server Update Step:** We build on the generalized FEDAVG formulation proposed by Reddi et al. [48], treating aggregated global parameter updates as an "antigradient" that can be input into different server optimizers (SGD is shown in Algorithm 1). Note that the server update operates on a weighted average of client updates as in McMahan et al. [44], weighted by $n_i = |\mathcal{D}_{i,q}|$.

We refer to the instantiation of this framework outlined here as FEDRECON below. We address frequently asked questions about FEDRECON and partially local federated learning in Appendix A.

## 4.1 Evaluation and Inference

To make predictions with global variables $g$ learned using Algorithm 1, clients can naturally reconstruct their local models just as they do during training, by using $R$, $g$, and $\mathcal{D}_{i,s}$ to produce local parameters $l$. Then $g$ and $l$ combined make up a fully trained partially local model, which can be evaluated on $\mathcal{D}_{i,q}$. We refer to this evaluation approach as RECONEVAL below. Note that this can be applied to clients unseen during training (most clients in large-scale settings), enabling inference for these clients.[2]

Reconstruction for inference is performed offline, independently of any federated process, so clients can perform reconstruction once and store local parameters for repeated use, optionally refreshing them periodically if they have new local data.

## 4.2 Connection to Meta Learning

Our framework is naturally motivated via meta learning. Given that RECONEVAL involves clients doing (gradient-based) reconstruction using global parameters, we ask: *Can we train global parameters conducive to fast reconstruction of local parameters?*

We can easily formulate this question in the language of model-agnostic meta learning [19]. The heterogeneous client distribution $\mathcal{P}$ corresponds to the heterogeneous distribution of tasks; each round (episode) we sample a batch of clients in the hope of meta-generalizing to unseen clients. Each client has a support dataset for reconstruction and a query dataset for global parameter updates. Our meta-parameters are $g$ and our task-specific parameters are $l_i$ for client $i$. We want to find $g$ minimizing the objective:

$$\mathbb{E}_{i \sim \mathcal{P}} \, f_i(g \parallel l_i) = \mathbb{E}_{i \sim \mathcal{P}} \, f_i(g \parallel R(\mathcal{D}_{i,s}, \mathcal{L}, g)] \tag{2}$$

---

[2]In this work we focus on new clients that have some local data for reconstruction; our method can be easily extended to learn a global default for the local parameters. We also show that skipping reconstruction can be reasonable for some tasks in Section 5.2.

where $g \parallel l_i$ denotes the concatenation of $g$ and $l_i$ and $f_i(g \parallel l_i) = \mathbb{E}_{\xi \in \mathcal{D}_{i,q}}[f_i(g \parallel l_i, \xi)]$.

In Appendix B we show that the instantiation of our framework where $R$ performs $k_r \geq 1$ steps of gradient descent on initialized local parameters using $\mathcal{D}_{i,s}$ and $U$ performs $k_u = 1$ step of gradient descent using $\mathcal{D}_{i,q}$ is *already* minimizing the first-order terms in this objective (*i.e.,* this version of FEDRECON is performing first-order meta learning). Intuitively, reconstruction corresponds to the MAML "inner loop" and the global parameter update corresponds to the "outer loop"; we test the same way we train (via reconstruction), a common pattern in meta learning.

Thus FEDRECON trains global parameters $g$ for fast reconstruction of local parameters $l$, enabling partially local federated learning without requiring clients to maintain state. In Section 5.2 we observe that our method empirically produces $g$ more conducive to fast, performant reconstruction on unseen clients than standard centralized or federated training (*e.g.,* see SERVER+RECONEVAL vs. FEDRECON in Table 1). We see in Figure 3 that *just one reconstruction step* is sufficient to recover the majority of performance.

# 5 Experimental Evaluation

## 5.1 Tasks and Methods

We next describe experiments validating FEDRECON on matrix factorization and next word prediction. We aim to determine whether reconstruction can enable practical partially local federated learning with fast personalization for new clients, including in settings without user-specific embeddings.

### 5.1.1 Matrix Factorization

We evaluate on federated matrix factorization using the popular MovieLens 1M collaborative filtering dataset [29]. We perform two kinds of evaluation:

1. STANDARDEVAL on *seen* users, those users who participated in at least one round of federated training. We split each user's ratings into 80% train, 10% validation, and 10% test by timestamp. We train on all users' train ratings, and report results on users test ratings.

2. RECONEVAL on *unseen* users, those users who did not participate at all during federated training. We split the users randomly into 80% train, 10% validation, and 10% test; we train with the train users and report results on test users.

The model learns $P$ and $Q$ such that $R \approx PQ^\top$ as discussed in Section 3, with embedding dimensionality $K = 50$. We apply FEDRECON with local user embeddings $P_u$ and global item matrix $Q$. We report root-mean-square-error (RMSE) and rating prediction accuracy. We compare centralized training, FEDAVG, and FEDRECON in Table 1. See also Appendix C.1 for more details on the dataset, model, and hyperparameter choices.

### 5.1.2 Next Word Prediction

We also aim to determine whether Federated Reconstruction can be successfully applied in settings without user-specific embeddings to improve robustness to client heterogeneity and communication cost, since our approach is agnostic to which parameters are chosen as local/global. We apply FEDRECON to next word prediction because the task provides a natural motivation for personalization: different clients often have highly heterogeneous data, *e.g.,* if they use different slang, but language models typically have a fixed vocabulary. We propose improving the ability of a language model to capture diverse inputs using local *out-of-vocabulary* (OOV) embeddings. OOV embeddings are a common application of the hashing trick [54] in deep learning; combining them with FEDRECON enables language models to effectively allow for personal input vocabularies for different clients. For example, if client $i$ frequently uses OOV token $t_i$ and client $j$ uses OOV token $t_j$, each client's corresponding local OOV embedding can learn to reflect this (even if the OOV embeddings collide). So adding local OOV embeddings with the core global vocabulary fixed can lead to improved personalization without more communication per round; we will also show that we can reduce the size of the core model (reducing communication) and get further benefits.

We perform next word prediction with the federated Stack Overflow dataset introduced in TensorFlow [51]. We use an LSTM model and process data similarly to Reddi et al. [48], comparing to their best

Table 1: Movielens matrix factorization root-mean-square-error (lower is better) and rating prediction accuracy (higher is better). STANDARDEVAL is on seen users, RECONEVAL is on held-out users. Results within 2% of best for each metric are in bold.

|  | RMSE ↓ | ACCURACY ↑ |
|---|---|---|
| CENTRALIZED + STANDARD EVAL | **.923** | **43.2** |
| CENTRALIZED + RECONEVAL | 1.36 | 40.8 |
| FEDAVG + STANDARD EVAL | .939 | 41.5 |
| FEDAVG + RECONEVAL | .934 | 40.0 |
| FEDRECON (OURS) | **.907** | **43.3** |

Table 2: Stack Overflow next word prediction accuracy and communication per round, per client. FEDYOGI and OOV/FULL FINETUNING require communication of all model parameters, FEDRECON does not (see Figure 2). Results within 2% of best for each vocabulary size are in bold.

| VOCAB. SIZE | 1K | 5K | 10K | COMMUNICATION |
|---|---|---|---|---|
| FEDYOGI | 24.3 | 26.3 | 26.7 | $2|l| + 2|g|$ |
| FEDRECON (1 OOV) | 24.1 | 26.2 | 26.4 | $\mathbf{2|g|}$ |
| FEDRECON (500 OOV) | 29.6 | 28.1 | 27.7 | $\mathbf{2|g|}$ |
| OOV FINETUNING (500 OOV) | 30.0 | 28.1 | 27.9 | $2|l| + 2|g|$ |
| FULL FINETUNING (500 OOV) | **30.8** | **29.2** | **28.8** | $2|l| + 2|g|$ |
| FEDRECON+FINETUNE (500 OOV) | **30.7** | **28.9** | **28.6** | $\mathbf{2|g|}$ |

FEDYOGI result. To demonstrate that reconstruction can be used to reduce model size, we describe experiments with vocabulary sizes [1000, 5000, 10,000]. See Appendix C.2 for details on the dataset, model, and hyperparameter choices.

## 5.2 Results and Discussion

In Tables 1 and 2 we present results for matrix factorization and next word prediction for FEDRECON and baselines. We call out several key comparisons below; more results can be found in Appendix D.

For the MovieLens task FEDRECON is able to match the performance of CENTRALIZED + STANDARD EVAL despite performing a more difficult task: as described in Section 5.1.1, FEDRECON is using RECONEVAL to evaluate on *held-out users*, reconstructing user embeddings for them and then evaluating. As is typical for server-trained matrix factorization models, CENTRALIZED + STANDARD EVAL is only being evaluated on *held-out ratings for seen users*. Note that we would not be able to evaluate on unseen users since they do not have trained user embeddings (randomly initializing them produces garbage results). If we reconstruct user embeddings for unseen users and then evaluate as in CENTRALIZED + RECONEVAL (we argue this is a fairer comparison with FEDRECON), we see that performance is significantly worse than FEDRECON and server-evaluation on seen users. One interesting finding was that the results of this seemed to vary widely across different users, with some users reconstructing embeddings no better than random initialization, while most others reconstructed better embeddings.[3] We see a similar result with FEDAVG for the MovieLens task, where FEDAVG with standard evaluation on seen users[4] performs a bit worse than CENTRALIZED + STANDARD EVAL, and performance for RECONEVAL on unseen users is significantly worse than FEDRECON. This indicates that FEDRECON is doing a better job of training global parameters *so they can reconstruct local parameters* than other approaches, as motivated in Section 4.2. Moreover, FEDRECON is doing this despite not having direct access to the data or the user-specific parameters–enabling this approach in settings where centralized training or FEDAVG is impossible.

---

[3]For this experiment, we repeat 500 times: sample 50 clients each time and perform RECONEVAL, reporting average metrics. Across runs, we observe large standard deviations of 1.7% accuracy (absolute) and 0.53 RMSE.

[4]Note that for this task, FEDAVG is equivalent in result to the stateful FEDPER approach in Arivazhagan et al. [4], since each client only updates its own user embedding. The user embeddings are stored on the server here, but this does not affect the result. Performance reduction may be caused by user embeddings getting "stale" across rounds which may occur when stateful algorithms are applied in cross-device FL, see Appendix A.

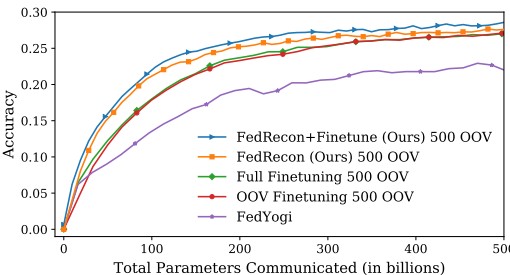

Figure 2: Accuracy as a function of total parameters communicated across all clients for FEDRECON and baselines for Stack Overflow next word prediction.

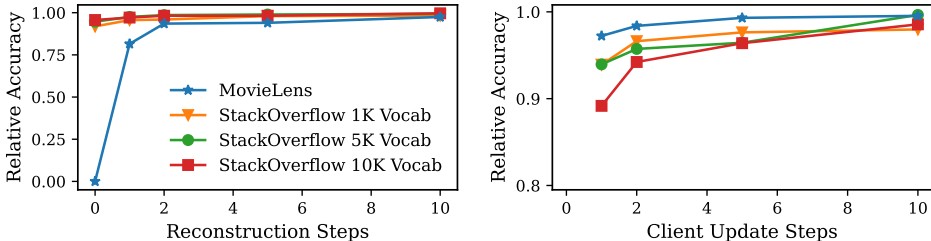

Figure 3: Accuracy compared to base FEDRECON when varying the number of reconstruction steps for local parameters (left plot) and client update steps for global parameters (right plot).

In the first section of the Stack Overflow results in Table 2, we compare FEDYOGI (an adaptive variant of FEDAVG introduced by Reddi et al. [48]) with FEDRECON, showing that enabling FEDRECON with 500 local OOV embeddings significantly boosts accuracy for every vocabulary size. Interestingly, we observe that accuracy actually improves for smaller vocabulary sizes for FEDRECON (500 OOV), whereas the reverse holds for FEDYOGI and FEDRECON (1 OOV). We posit that this is because decreasing the vocabulary size effectively increases the amount of "training data" available for the local part of the model, since OOV embeddings are only used (and trained) when tokens are out-of-vocabulary; this is useful only when the local part of the model has sufficient capacity via the number of OOV embeddings. This hypothesis is consistent with vocabulary coverage: a 10K vocabulary covers 86.9% of the tokens in the dataset, a 5K vocabulary covers 80.1%, and a 1K vocabulary covers 49.2%; we see that difference in results for FEDRECON (500 OOV) is greater between 1K and 5K than between 5K and 10K. We caution that reducing vocabulary size may be undesirable in some cases: reducing the size of the vocabulary also restricts the output tokens of the model.

**Comparing with Finetuning:** In Table 2 we compare FEDRECON with FINETUNING [52, 58] to study whether reconstruction can provide similar benefits as global personalization methods. In our implementation we train a fully global model using FEDYOGI, perform local gradient steps to finetune part of the model using the support set, and then evaluate on the query set (same sets as used for FEDRECON). For OOV FINETUNING, the OOV parameters only are finetuned using the support set (comparable to FEDRECON), and for FULL FINETUNING all parameters are finetuned. Comparing FEDRECON (500 OOV) and OOV FINETUNING, we see that reconstructing local embeddings performs similarly to finetuning pre-trained OOV embeddings, despite FEDRECON not communicating the local parameters $l$ to the server. FULL FINETUNING from Table 2 achieves better accuracy since all parameters are finetuned. To compare this fairly with reconstruction, we perform FEDRECON+FINETUNE, where the support set is used first to reconstruct local parameters and then to finetune global parameters before evaluation. We also see that we can get comparable results, indicating that reconstruction can enable personalization on (potentially privacy-sensitive) local parameters while reducing communication. See Figure 2 for a comparison of different approaches by the total number of parameters communicated–we see an advantage for FEDRECON, particularly for lower total communication.

**Varying Reconstruction Steps:** In Section 4.2 we described a connection between our framework and MAML [19], which has been a successful paradigm for fast adaptation to tasks with few steps. In Figure 3 we perform FEDRECON for varying numbers of reconstruction steps $k_r \in [0, 1, 2, 5, 10]$ and plot the accuracy as a fraction of accuracy across tasks from Tables 1 and 2. We see that

for zero reconstruction steps (an ablation skipping reconstruction), MovieLens accuracy is 0.0, as expected (all user embeddings are randomly initialized). Relative accuracy for Stack Overflow NWP settings remains above 90%, suggesting that for this task clients can still perform inference with a FEDRECON-trained model even without any data to reconstruct. Importantly, *just one reconstruction step* is required to recover the majority of remaining performance across both tasks, indicating that FEDRECON learns global parameters conducive to fast reconstruction.

**Varying Client Update Steps:** In Section 4.2 we showed that gradient-based FEDRECON, involving $k_r \geq 1$ reconstruction steps and $k_u = 1$ client update steps, is minimizing a first-order meta learning objective for training global parameters that yield good reconstructions. In Figure 3 we perform FEDRECON with $k_u \in [1, 2, 5, 10]$ and compute relative accuracy as a fraction of accuracy across tasks from Tables 1 and 2. For each experiment we run for a fixed number of rounds. We see that 1 step recovers almost all of the accuracy and adding more steps gradually increases accuracy further. Interestingly, we observe that for $k_u = 1$ training proceeds significantly slower than for other values such that performance is still slightly increasing after the fixed number of rounds. This is analogous to the difference between FEDAVG and FEDSGD [44]. While FEDSGD is optimizing the original learning objective, FEDAVG often achieves similar performance in significantly fewer rounds by adding multiple gradient steps on aggregated parameters.

We present further baselines and ablations in Appendix D.

# 6    Open-Source Library

We are releasing a code framework for expressing and evaluating practical partially local federated models built on the popular TensorFlow Federated library [50]. The code is released under Apache License 2.0. In addition to allowing for easy reproduction of our experiments, the framework provides a flexible, well-documented interface for researchers and modelers to run simulations in this setting with models and tasks of their choice. Users can take any existing Keras model and plug it into this framework with just a few lines of code. We provide libraries for training and evaluation for MovieLens matrix factorization and Stack Overflow next word prediction, which can be easily extended for new tasks. We hope that the release of this framework spurs further research and lowers the barrier to more practical applications.

# 7    Deployment in a Mobile Keyboard Application

A key differentiator of our method is that it scales to practical training and inference in cross-device settings with large populations. To validate this, we deployed FEDRECON to a mobile keyboard application with hundreds of millions of federated learning clients. We used a system similar to Bonawitz et al. [7] to deploy FEDRECON for training. Note that the system does not support stateful clients given the issues with large-scale stateful training described in Section 4, so a stateless approach was necessary for deployment.

Users of the mobile keyboard application often use *expressions* (GIFs, stickers) to communicate with others in *e.g.,* chat applications. Different users are highly heterogeneous in the style of expressions they use, which makes the problem a natural fit for collaborative filtering to predict new expressions a user might want to share. We trained matrix factorization models as described in Section 5.1.1, where the number of items ranged from hundreds to tens of thousands depending on the type of expression.

Training in production brought challenges due to data sparsity. Depending on the task, some clients had very few examples, if *e.g.,* they didn't commonly share stickers via the keyboard application. To ensure clients with just one example weren't just adding noise to the training process by participating, we oversampled clients and filtered out the contributions of clients without at least some number of examples. We reused examples between the support and query sets as described in Appendix D to ensure all examples were used for both reconstruction and global updates.

Another practical challenge we faced was orthogonal to our method and commonly faced in real-world federated learning applications: heterogeneity in client resources and availability meant that some participating clients would drop out before sending updates to the server. We found that the simple strategy of oversampling clients and neglecting updates from dropped-out clients appeared to

perform well, but we believe studying the fairness implications of this is a valuable area for future work.

After successful training, the resulting model was deployed for inference in predicting potential new expressions a user might share, which led to an increase of 29.3% in click-through-rate for expression recommendations. We hope that this successful deployment of FEDRECON demonstrates the practicality of our approach and leads the way for further real-world applications.

## 8 Conclusion

We introduced Federated Reconstruction, a model-agnostic framework for fast partially local federated learning suitable for training and inference at scale. We justified FEDRECON via a connection to meta learning and empirically validated the algorithm for collaborative filtering and next message prediction, showing that it can improve performance on unseen clients and enable fast personalization with less communication. We also released an open-source library for partially local federated learning and described a successful production deployment. Future work may explore the optimal balance of local and global parameters and the application of differential privacy to global parameters (see Appendix E).

## Acknowledgments and Disclosure of Funding

We thank Brendan McMahan, Lin Ning, Zachary Charles, Warren Morningstar, Daniel Ramage, Jakub Konečný, Blaise Agüera y Arcas, and Jay Yagnik from Google Research for their helpful comments and discussions. We also thank Wei Li, Matt Newton, and Yang Lu for their collaboration towards deployment.

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
