# OpenReview forum: "Federated Reconstruction: Partially Local Federated Learning"
_NeurIPS.cc/2021/Conference — NeurIPS 2021 Poster_

### Official Review · Reviewer_fqwY · 2021-07-05

**Rating:** 6
**Confidence:** 3

**Summary:**

This paper introduces a partially local federated learning method named FEDRECON based on federated reconstruction.  The experimental results demonstrate the effectiveness of FEDRECON. The authors also deploy this approach in a real-world mobile keyboard application.

**Limitations And Societal Impact:**

The authors discussed the privacy-related issues of the proposed approach and related limitations in the appendix. However, it is encouraged to include some discussions in the main content.

**Main Review:**

Pros:
1. This paper is well written and easy to follow.
2. The proposed approach is interesting.
3. The authors provide a nice code library.

Cons:
1. The contributions of this work over other partially local federated learning methods are not clearly explained.
2. The motivation of reconstruction is not clarified.
3. Some related works are not cited nor compared.

Detailed comments:
1. The technical contributions of this work are not well elaborated. Partially local federated learning is a widely studied problem [1-5], especially in the recommendation field. In these methods, each client locally learns a user embedding while item embeddings are global parameters. The authors' discussions in Appendix A do not clarify why this setting is not optimal.
2. The motivation of reconstruction is also not clarified. For example, in federated recommender systems, why personalized parameters can be generated from shared global ones? What is the intuition?
3. Some related works are not cited nor compared. The authors need to compare the methods in the recommendation field that locally keep personalized parameters. In addition, the authors may consider comparing some meta-learning based federated learning methods like [2] to better support the key statements.

[1] Chai, Di, et al. "Secure federated matrix factorization." IEEE Intelligent Systems (2020).
[2] Lin, Yujie, et al. "Meta Matrix Factorization for Federated Rating Predictions." SIGIR 2020.
[3] Ge, Suyu, et al. "Fedner: Privacy-preserving medical named entity recognition with federated learning." arXiv preprint arXiv:2003.09288 (2020).
[4] Ammad-Ud-Din, Muhammad, et al. "Federated collaborative filtering for privacy-preserving personalized recommendation system." arXiv preprint arXiv:1901.09888 (2019).
[5] Wu, Chuhan, et al. "Fedgnn: Federated graph neural network for privacy-preserving recommendation." arXiv preprint arXiv:2102.04925 (2021).

**Time Spent Reviewing:**

3 hours

---

> ### Author Response · Authors · 2021-08-10
> **Regarding motivation and relationship to previous work**
>
> We thank the reviewer for their time and effort, and for their valuable suggestions for improving the clarity of the paper. We’re glad to see the reviewer thought our paper was well-written and appreciated our open-source code library and real-world deployment.
>
> We did want to clarify the core motivation of our work and relationship to previous work:
> 1. “The contributions of this work over other partially local federated learning methods are not clearly explained.” Our work is the first *stateless* general method for partially local FL. See the first question in Appendix A (FAQ) and Sections 1 and 3 for motivation for statelessness, but in short: in a large-scale cross-device setting, client state will rarely be available and may become stale, degrading performance (see the discussion on SCAFFOLD in [6] Sec. 5.1). We observe this degradation empirically in our work by comparing against the *stateful* FedPer method [7]. Importantly, the real-world deployment described in Section 7 would not be possible with a stateful method.\
> \
> Previous methods in partially local federated learning mentioned by the reviewer, including [1], [3], [4], and [5], are stateful. As we observed by comparing against FedPer [7], which is essentially a generalization of stateful matrix factorization methods, this results in degradation in the cross-device setting. [2] applies meta learning for federated matrix factorization, but is not partially local since gradients for user parameters are sent to the server (which can cause privacy concerns, as the authors discuss in [2] Sec. 7). We’ve added further discussion of these related works in Section 2 (which already discusses [1], [2], [4]) to clarify these differences.
> 1. “The motivation of reconstruction is also not clarified… why personalized parameters can be generated from shared global ones?” In Section 4.2 and Appendix B we provide intuitive and theoretical arguments for why reconstruction naturally leads to training global parameters that can be easily used to train personal local parameters, similar to the existing intuition around Model-Agnostic Meta Learning methods [8]. For matrix factorization in particular, alternating minimization methods (where personal parameters are trained based on frozen global ones) are well-established and quite successful [9]. In Figure 3 and Table 1, we also empirically demonstrate that reconstruction leads to performant final models with minimal gradient steps (even on unseen clients). In the FAQ in Appendix A (second question), we also summarize other motivations for reconstruction (more details in Sections 1, 3, and 4): producing local models offline for inference, enabling production applications like that described in Section 7; lack of significant additional computational cost; reduction in communication cost; and avoiding staleness issues since local parameters are generated on-the-fly. To clarify these points, we’ve modified the FAQ to add more discussion about the core motivation for reconstruction.
> 1. We discuss the works mentioned by the reviewer in comment (1). To summarize: previous partially local FL methods are stateful, and we do compare against FedPer [7], a representative stateful method that generalizes some of the mentioned works. We observe performance degradation when applying this method in cross-device FL due to the staleness issue observed for SCAFFOLD in [6] Sec. 5.1. [2] involves sending user-specific gradients to the server, which can cause privacy concerns as discussed in [2] Sec. 7. [2] is also specific to the matrix factorization application, whereas our method is agnostic to the model and task as we show in the next word prediction experiments.
>
> Given these clarifications, particularly around core motivation for the work, we hope the reviewer will consider adjusting their initial rating.
>
> ---
>
> [1] Chai, Di, et al. "Secure federated matrix factorization." IEEE Intelligent Systems (2020).
>
> [2] Lin, Yujie, et al. "Meta Matrix Factorization for Federated Rating Predictions." SIGIR 2020.
>
> [3] Ge, Suyu, et al. "Fedner: Privacy-preserving medical named entity recognition with federated learning." arXiv preprint arXiv:2003.09288 (2020).
>
> [4] Ammad-Ud-Din, Muhammad, et al. "Federated collaborative filtering for privacy-preserving personalized recommendation system." arXiv preprint arXiv:1901.09888 (2019).
>
> [5] Wu, Chuhan, et al. "Fedgnn: Federated graph neural network for privacy-preserving recommendation." arXiv preprint arXiv:2102.04925 (2021).
>
> [6] "Adaptive Federated Optimization": https://arxiv.org/abs/2003.00295
>
> [7] “Federated Learning with Personalization Layers”: https://arxiv.org/abs/1912.00818
>
> [8] “Model-Agnostic Meta-Learning for Fast Adaptation of Deep Networks”: https://arxiv.org/abs/1703.03400
>
> [9] “Low-rank Matrix Completion using Alternating Minimization”: https://arxiv.org/pdf/1212.0467.pdf

---

### Official Review · Reviewer_3vKL · 2021-07-14

**Rating:** 7
**Confidence:** 4

**Summary:**

The authors propose a new approach to partially local federated learning. In contrast to previous works their approach does not require agents to maintain state (i.e. store the local parameters of the model). Instead, they reconstruct the personalized parts of the model on the fly from local data.  Such a state-less approach is crucial for any practical deployment of FL at scale since is enables new clients that have not yet trained their local parameters to participate in training and to perform inference.
Their approach (FedRecon) is inspired by model-agnostic meta learning, where fast reconstruction at test-time is a key design criteria.

Empirically the performance of FedRecon is evaluated along different axis. The authors show that their approach trains global parameter that are particularly well suited for reconstruction, the approach achieves performance comparable to personalization through fine-tuning of a global model, and they study the qualitative behavior of their method wrt several hyperparameters.
In addition the authors report on a successful real-world deployment of the method across hundreds of millions of clients and discuss the practical challenge of data sparsity.

The code for reconstructing the experiments is packaged as a library to allow users to seamlessly integrate other tasks and algorithms.


**Limitations And Societal Impact:**

yes

**Main Review:**

The paper extends partially local federated learning to settings where agent can not maintain state. This scenario seems well motivated from a practical perspective and the authors give several good application examples. The core idea of using a local reconstruction algorithm to obtain the local parts of the model is simple but interesting and to the best of my knowledge it is novel in this context.

My main criticism is the presentation of the empirical results in Section 5. The baselines are not well introduced and the goal of the individual comparisons is not clear a priori. This makes it hard to follow (in particular the first part in 5.2, Table 1/2) and as a result your baselines seem a bit constructed at a first glance.

I think it would help if you make clear from the beginning that a direct comparison to existing methods is not possible because there is no other state-less approach that keeps parts of the model parameters entirely local. Then, you could one-by-one outline what aspect of your method you want to study and then introduce the baselines for this purpose. For example, the ReconEval evaluation scheme for Table 1 would make more sense to the reader if they knew that you aim to show that your method learns global parameters that are particularly well suited for reconstruction later on, in comparison to other methods of obtaining these global parameters.

In this context, a comparison to a global only model might also be interesting, i.e., a model where you ignore the private features completely. This would help clarify the need for local steps in the ReconEval procedure that you introduce.

In 263 you mention that results vary a lot across users. Could you elaborate more on this? Showing the distribution over accuracy values of users for the different methods (instead of only RMSE) would be useful here and provide more insights into this interesting finding.

It is nice to see that you actually deployed this federated learning method in a real application. You mentioned that data sparsity was an issue and prevented the naïve implementation of your method.  Were there any other practical challenges you were facing? You mention a 29.3% increase in CTR, this claim is not particularly useful without a reference. What method was used in the previous system?

Overall, I think the authors adequately addressed the concerns of the meta-review from the previous review round. If the authors are willing to make an effort to address my concerns about the presentation of the empirical results, I will recommend acceptance.


**Time Spent Reviewing:**

4

---

> ### Author Response · Authors · 2021-08-10
> **We’ve addressed the reviewer’s valuable comments**
>
> We thank the reviewer for their time and effort. We’re encouraged to see that the reviewer appreciated the motivation for stateless partially local FL and the novelty of our approach. The reviewer also appreciated changes we made after our ICML meta review (reproduced in full in the submission), our real-world deployment, and our open-source library.
>
> The reviewer made excellent suggestions around improving the clarity of our presentation of results. We’ve addressed them:
> * The reviewer is correct in that there cannot be a completely direct comparison to existing methods, since no other general method for stateless partially local FL exists. We’ve made changes in Section 5.1 and 5.2 to clearly enumerate baselines and make clear what aspect of performance is being tested by each baseline (performance on seen clients, unseen clients, communication tradeoffs, etc.).
> * Responding to: “a comparison to a global only model might also be interesting, i.e., a model where you ignore the private features completely” In Figure 3, we do run experiments for both MovieLens and Stack Overflow where we skip the local reconstruction steps entirely, i.e. ignoring the local features. As mentioned by the reviewer, this is an ablation studying the necessity of reconstruction steps, and we see performance degradation for both matrix factorization and next word prediction tasks, as expected. We’ve updated the discussion in Section 5.2 to emphasize this result.
> * We’ve also added more details on hyperparameters used in baselines in Appendix C. This is in addition to the documentation and default values for all training hyperparameters present in the open-source code library.
> * “In 263 you mention that results vary a lot across users. Could you elaborate more on this?” We’ve run an additional experiment for this Server+ReconEval setting measuring sample standard deviation of accuracy and RMSE across 500 runs. As mentioned in lines 263-265, the results in this setting had worse average performance and relatively high variance across runs, providing empirical support for our claim that FedRecon leads to better reconstruction performance than standard server training. We now quantify this in the paper, observing large standard deviations of 1.7% (absolute) for accuracy and 0.53 RMSE (mean accuracy 40.8, RMSE 1.36).
> * “It is nice to see that you actually deployed this federated learning method in a real application. You mentioned that data sparsity was an issue... Were there any other practical challenges you were facing?” Another practical challenge we faced was orthogonal to our method and commonly faced in real-world federated learning applications: heterogeneity in client resources and availability meant that some participating clients would drop out before sending updates to the server. We found that the simple strategy of oversampling clients and neglecting updates from dropped-out clients appeared to perform well, but we believe studying the fairness implications of this is a valuable area for future work. We’ve added discussion of this in Section 7.
>
> We thank the reviewer again for their comments–they have significantly strengthened our work.

---

> > ### Comment · Reviewer_3vKL · 2021-08-19
> > **Thanks for addressing the comments**
> >
> > I appreciate that the authors addressed and incorporated the comments by me and by the other reviewers.
> > In particular, the suggestions related to the presentation of the experimental section.
> >
> > One question I am still curious about - in your practical deployment in the mobile keyboard application you claim a 29.3% increase in CTR, but you do not mention what the baseline was.
> > What was the algorithm used in the reference implementation? Was the previous approach also stateless (you mention a stateful approach is not possible)? What were the key innovations that lead to this significant improvement?
> >
> > Thanks for clarifying

---

> > > ### Author Response · Authors · 2021-08-20
> > > **Thanks and clarification on the deployment**
> > >
> > > We're glad to hear that the reviewer appreciated our clarifications and incorporation of reviewer feedback.
> > >
> > > For the previous method used in the mobile keyboard application: because there was previously no stateless partially local FL method that could be used for matrix factorization in this large-scale cross-device FL setting, a simpler (stateless) approach was used. Distributed privacy-preserving analytics was used to count co-occurrences of pairs of expressions (e.g. stickers, GIFs) used by the same user within a fixed vocabulary. These co-occurrences were summed across users. This yielded a table of item-item co-occurrences on the server. Item similarity was calculated by calculating the Jaccard index between different items in this table. At recommendation time, the item with the highest average item similarity with the items previously used by the user was recommended to the user. The 29.3% relative improvement was observed in an A/B live experiment in the mobile keyboard application comparing our method with the previous method.
> > >
> > > Our hypotheses for reasons for the improvement:
> > > 1. Matrix factorization methods, newly enabled in this setting, tend to outperform the approach described above.
> > > 2. Our method makes use of more granular data, i.e. the raw user-item co-occurrences rather than just the aggregated item-item ones.
> > > 3. Federated Reconstruction trains for improved reconstruction performance on unseen users (see the improved performance vs. server matrix factorization on unseen users in Table 1), which is important in this setting because the vast majority (99%+) of users are unseen during training.

---

> > > > ### Comment · Reviewer_3vKL · 2021-08-27
> > > > **Thank you for the details**
> > > >
> > > > Thank you for providing more details on the evaluation of your deployment. It would be valuable for the reader if, in a final version, you could also elaborate more on this baseline and the role of the statelessness and model reconstruction at the example of this use-case. Without this reference the 29.3% improvement claim is not particularly useful.
> > > > Overall I will stay with my score, I think it is a good paper.

---

### Official Review · Reviewer_fiWt · 2021-07-16

**Rating:** 6
**Confidence:** 5

**Summary:**

In this paper, the authors proposed a new FL framework to address data privacy, communication cost, and non-iid data distribution issues in FL. In the proposed method, clients only send and receives partial models to cloud and reconstruct the other half from the clients' own data.

**Main Review:**

I have the following major concerns about the paper.

1. What does the model agnostics refer to? Do the authors mean the clients can have different models or the framework can handle different tasks? Please clarify.

2. For the tasks, I agree the task like matrix factorization is suitable for the proposed method. However, for other tasks where each client trains a deep model and aggregates all the clients' models on the cloud, I don't think only compare with FedYogi is enough. FedYogi is an extension from FedAvg which in general only produces a global model and is not suitable for personalization. In my opinion, the authors should compare at least of the SOTA personalized FL methods for their performance, such as

a. FedBN: Federated Learning on Non-IID Features via Local Batch Normalization

b. Personalized Cross-Silo Federated Learning on Non-IID Data

c. Federated Optimization in Heterogeneous Networks

d. SCAFFOLD: Stochastic Controlled Averaging for Federated Learning

e. Adaptive Personalized Federated Learning

f. Federated learning ´of a mixture of global and local models

3. The proposed method is a heuristic method without any theoretical convergence guarantee.

4. I also wonder whether this framework works if the local model is a complex and large model. In addition, how does the partition size affect the proposed framework? Especially, a large model is needed.


**Time Spent Reviewing:**

2

---

> ### Author Response · Authors · 2021-08-10
> **Title: Regarding differences from previous works**
>
> We thank the reviewer for their time and effort, and for their valuable suggestions and questions concerning the paper. We address their numbered comments below:
>
> 1. “Model-agnostic” refers to the framework being generic to the task and model, i.e., any model can be split into global and local parts and trained using FedRecon in principle, even models without naturally user-specific parameters. We’ve revised the presentation in Section 3 to clarify that the framework does not involve clients updating different models.
> 1. We thank the reviewer for taking the time to point out related works. We'd like to highlight that in this paper we focus on the cross-device FL setting, characterized by partial participation and stateless clients (a, b, d, e, f all require stateful clients). A crucial contribution of our method is enabling *stateless* partially local federated learning. See the first question in Appendix A (FAQ) and Sections 1 and 3 for motivation for statelessness, but in short: in a large-scale cross-device setting, client state will rarely be available and may become stale, degrading performance (see [2] Sec. 5.1 for a discussion in context of SCAFFOLD). We observe this degradation empirically in our work by comparing against the *stateful* FedPer method. Importantly, the real-world deployment described in Section 7 would not be possible with a stateful method.\
> \
> Moreover, the mentioned works mostly do not address the problem of partially local federated learning (stateless or not). In this setting, some (privacy-sensitive) parameters are never aggregated by the server (c, d learn a fully global model similar to FedYogi; b, e, f communicate all model parameters to the server; (a) is partially local but the method is stateful and specific to batchnorm parameters). For example, if FedProx were directly applied to the matrix factorization setting, individual user embedding updates would be sent to the server, trivially leaking user preferences. Note that in addition to methods producing fully global models and the stateful partially local FL method FedPer, we do compare to finetuning methods for personalization in Figure 2 and Table 2, demonstrating that FedRecon has an improved accuracy-communication tradeoff.\
> \
> See below table for a more detailed comparison of FedRecon and other mentioned works. We’ve updated related work in Section 2 to more clearly distinguish our work from these works.
> 1. We show that our approach converges empirically in Figures 2, 4, and 5. We also show in Table 1 that we achieve better results on unseen users than a **server-trained** matrix factorization model, again demonstrating empirical performance. We also show a theoretical connection to MAML in Appendix B, which justifies the fast reconstruction in Figure 3. While we do not present additional theoretical convergence results, one of our core contributions in this work is our deployment of this approach at scale in a real-world setting with hundreds of millions of clients, improving recommendation CTR by 29.3% (see Section 7). We believe that this contribution does more to validate the approach than theoretical convergence results.\
> \
> A note on the research challenge of proving convergence guarantees in this space, which is an interesting area for future work: standard proving techniques for matrix factorization via alternating minimization (such as [1]) rely on: 1) a good initialization point of the user/item embeddings (usually constructed from SVD); 2) the optimal (or close-to-optimal) user (or item) embeddings can be obtained given the item (or user) embeddings. These two conditions are difficult to meet in the federated setting, where the ratings are distributed across the users, and each user performs training over multiple local epochs with local data (as in FedAvg). Proving convergence guarantees for alternating minimization algorithms in the federated setting would require new proving techniques (e.g., [2]), which is itself an interesting future research direction.
> 1. Responding to: “In addition, how does the partition size affect the proposed framework?” We do present experiments with differing splits of local and global parameters in Tables 2 and 4, Figures 3, 5, and 6. In particular, Figure 6 presents results with ten different partitions. On evaluation with large models, the Stack Overflow next word prediction model has up to 4.2M parameters, with up to 960,000 local parameters per client. To our knowledge, this model is one of the largest models trained in partially local federated learning work (and larger than the models in most of the above mentioned work). More broadly, in cross-device FL, training of larger models remains an open research and systems challenge due to client resource heterogeneity.
>
> We categorize the works mentioned by the reviewer in further detail:
>
> |                 | Stateless | Fully Global Model | Cross-Device | Matrix Factorization | Server Has Local Models |
> |-----------------|-----------|--------------------|--------------|----------------------|-------------------------|
> | FedRecon (Ours) | Y         | N                  | Y            | Y                    | N                       |
> | FedBN [3]       | N         | N                  | Y            | N                    | N                       |
> | FedAMP [4]      | N         | N                  | N            | N                    | Y                       |
> | FedProx [5]     | Y         | Y                  | Y            | N                    | N/A                     |
> | SCAFFOLD [6]    | N         | Y                  | N            | N                    | N/A                     |
> | APFL [7]        | N         | N                  | N            | N                    | N                       |
> | L2GD [8]        | N         | N                  | Y            | N                    | Y                       |
>
> Explanation of column headers:
> * Stateless: whether clients need to maintain state. See first question of FAQ (Appendix A) for the importance of statelessness in cross-device FL. The real-world application described in Section 7 would not be possible with a stateful method.
> * Fully Global Model: whether the method learns only a single global model instead of personalized models.
> * Cross-Device: whether the method is intended for use in large-scale cross-device FL, with large population sizes (e.g. 1M+) and partial participation (e.g. 500 clients per round). Stateful methods intended for cross-device use may degrade in performance.
> * Matrix Factorization: whether the method can be applied to matrix factorization, where sending user embeddings (or their updates) to the server trivially leaks user preferences. This is an intuitive check for whether the related work explores general partially local FL. Of the mentioned works, only FedRecon is appropriate for matrix factorization.
> * Server Has Local Models: whether the server needs access to the raw updates / models of individual clients (rather than their sum or average). This can be problematic due to privacy limitations.
>
> Given these clarifications, particularly around related work and our experiments, we hope the reviewer will consider adjusting their initial rating.
>
> ---
>
> [1] "Low-rank Matrix Completion using Alternating Minimization": https://arxiv.org/abs/1212.0467
>
> [2] "Adaptive Federated Optimization": https://arxiv.org/abs/2003.00295
>
> [3] “FedBN: Federated Learning on Non-IID Features via Local Batch Normalization”: https://arxiv.org/abs/2102.07623
>
> [4] “Personalized Cross-Silo Federated Learning on Non-IID Data”: https://arxiv.org/abs/2007.03797
>
> [5] “Federated Optimization in Heterogeneous Networks”: https://arxiv.org/abs/1812.06127
>
> [6] “SCAFFOLD: Stochastic Controlled Averaging for Federated Learning”: https://arxiv.org/abs/1910.06378
>
> [7] “Adaptive Personalized Federated Learning”: https://arxiv.org/abs/2003.13461
>
> [8] “Federated Learning of a Mixture of Global and Local Models”: https://arxiv.org/abs/2002.05516

---

> > ### Comment · Reviewer_fiWt · 2021-08-19
> > **Thanks for addressing the comments**
> >
> > The authors have addressed my most concerns. But I still believe it will be nice to have some theoretical results of the proposed algorithm. Thus, I have increased one point of my score.

---

### Official Review · Reviewer_TSbD · 2021-07-22

**Rating:** 7
**Confidence:** 4

**Summary:**

- This work proposed FedRecon, a method that splits the client model into local and global parameters to enable stateless federated learning
- During training, the client trains the local and the global model using separate splits of the client dataset, but only transmit the global data for aggregation
- Clients with no local parameters trained (clients not seen during the training), can reconstruct/train their local parameters after fetching the global parameters from the server using the local training data. Similarly, other clients which might have either stale local parameters or might have lost/forgotten the local parameters can reconstruct them too
- This makes the method stateless since the clients need not store the global parameters (sever stores them) and the local parameters (since they can be reconstructed)
- The authors show good empirical performance on matrix factorization and next word prediction task. Additionally, the authors also the practical utility and scalability of the method through the deployment of the method in a real-life scenario.

**Limitations And Societal Impact:**

N.A.

**Main Review:**

**Strengths**

- The proposed method is a simple extension of the standard federated learning setup, with the difference being only in how the local model is trained. Experiments show good performance with a wide range of hyperparameters (support and query data splits, local update steps, etc), and the method is applicable to a wide array of models.
- I also appreciate the authors providing the prior submission review. It helped me understand how the paper has evolved over multiple submission cycles.
- Open-sourcing of code through the TensorFlow Federated Library for future experimentation and practical use is a good contribution. However, it might be my ignorance of the framework but I would like to understand how is it different from someone open-sourcing the code for any other submission to the conference. Is the open-sourced code the same used by authors to run their experiments, or they modified it especially to release it as a separate framework?
- The deployment to a mobile application proves the practical applicability of the method and is a good contribution. However, the authors mention in the Submission History section that "[practical deployment results are] rare in FL papers which typically achieve only simulation results". I would disagree with this comment. Results on simulated settings do not necessarily mean inapplicability to practical deployments. Recent datasets do have clients in the order of hundreds of thousands, which mimics the practical setting to a certain extent. Additionally, very few researchers have the resources to deploy to millions of devices, and while these results are good to have in the paper, their absence should by no means be negatively looked at.


**Clarifications and Improvements**

- Table 1: What is the Server + Standard Eval and Server + ReconEval training setup? Can the authors describe this somewhere in the paper or refer to a prior work that utilizes this setup?
- Table 1, 2: FedRecon for matrix factorization is trained for 50 gradient steps each for the local and global params, and for NWP it's trained for 100 gradient steps each. How many gradient steps were the baselines trained for? Can the authors please provide the hyperparameters used to train the baselines as well?
- Lines 263 - 265: The authors mention that the results for Server+ReconEval varied widely across users. Since they ran 500 iterations of the experiment, can the authors please provide the standard error numbers along with the reported mean accuracy to help better understand the variation amongst clients, and compare this variation with FedRecon
- I felt that the experiment section was informative but was slightly convoluted to follow and understand. For a future version of the paper, the authors might consider refactoring the section to make it more readable.



Overall, I felt this work:

- tackles a relevant problem in federated learning,
- proposes a simple, widely applicable to solve the problem, and
- empirically shows good performance in simulated and practical scenarios.

Thus, I would vote for its acceptance.

**Time Spent Reviewing:**

4

---

> ### Author Response · Authors · 2021-08-10
> **Thank you; Clarifications and improvements**
>
> We thank the reviewer for their time and effort, and for their valuable suggestions for improving the clarity of the paper. We are glad to see that the reviewer finds our work to be widely applicable and performant in both simulation and real-world scenarios. The reviewer also appreciated our description of substantial changes since our ICML meta-review.
>
> We have addressed each suggestion in order:
> * We revised and enumerated the descriptions for StandardEval and ReconEval in Section 5.1.1, clarifying the evaluation setup.
> * We added additional details about hyperparameters used for the baselines in Appendix C. We also want to highlight that our open-source code library also thoroughly documents the default hyperparameters used for different settings, enabling easy reproduction of our experiments.
> * We ran the reviewer’s suggested experiment measuring sample standard deviation of matrix factorization metrics for Server+ReconEval. As mentioned in lines 263-265, the results in this setting had worse average performance and relatively high variance across runs, providing empirical support for the claim that FedRecon leads to better reconstruction performance than standard server training. We now quantify this in the paper, observing large standard deviations of 1.7% (absolute) for accuracy and 0.53 RMSE (mean accuracy 40.8, RMSE 1.36).
> * We have modified the labels for different evaluation settings in Section 5.2 to use more consistent terminology, making the section easier to follow.
>
> To answer the reviewer’s other question: “they modified it especially to release it as a separate framework?” Yes, we have modified the code so that the core framework (independent of tasks like matrix factorization or training loop code) is available as a generic interface, similar to the TensorFlow Federated interface for using Federated Averaging commonly used by researchers. We hope that this contribution will accelerate further research and applications in this space.
>
> We thank the reviewer again for their comments–our work is stronger as a result.

---

> > ### Comment · Reviewer_TSbD · 2021-08-25
> > **Response to authors rebuttal**
> >
> > Thank you authors for your response. I'm glad to see that my review helped make your work better. I'll keep my score as it is, for I feel that it is appropriate for this work.

---

### Decision · Program_Chairs · 2021-09-27

**Decision:**

Accept (Poster)

**Comment:**

I have carefully read all reviews, rebuttals and subsequent discussion. The proposed methodology relies on a sequence of intuitive shortcuts, reformulations and claims, which in totality make intuitive sense, but are not supported by any theory, and as such, the approach remains a heuristic. For example, the original cross device FL formulation is not well motivated, despite the fact that it is used in nearly every paper on the topic. This is not a particular issue with the current paper - this is an issue with all works in this area.  The authors rightfully claim that statelessness can be a problem in practical deployments of FL in the cross device setting. However, statefulness, when present, has a clear purpose: well designed stateful methods can achieve superior theoretical guarantees, and this is the main reason why states are useful. If the authors tested again stateful methods in the regime when states are useful (e.g., states in SCAFFOLD help to reduce client drift, which leads to better theoretical rates in certain regimes; there are dozens of examples like this), I doubt their stateless approach could win. Of course, once a method is applied outside of the scope in which it is theoretically shown to be better than competing approaches, all bets are off, and the empirical behavior can go both ways. I appreciate though that in the setting considered by the authors, storing states is not practical. The authors then proceed to a reformulation of the cross device FL problem in the form of (2), but do not make any attempt to explain what is lost by using this reformulation. Clearly, solving (1) and (2) are two very different tasks. The connection provided between (2) and (1) is intuitive only. I believe one can think of simple counterexamples that show that by solving (2), one does not approach the solution of (1) in any reasonable way. In this sense, their reformulation is a heuristic again. It may work in some regimes, but it is not clear what regimes these are. Some theory is sorely needed here. In totality, the authors provide a heuristic which may or may not work, depending on the problem being solved. This reduces its usefulness / applicability in regimes/applications beyond the ones studied in the paper. Having said that, the approach seems to be useful for the application at hand. The connection of (2) to MAML is helpful. However, outlining this connection alone does not mean that by solving (2) one gets any close to solving (1). In that sense, the authors could have well just started by saying that what they want to solve is (2) to start with. However, why should one formalize a FL problem in the form (2)? Say we can solve this problem exactly. What does it say about the generalization properties of the model we learn that way? Why does this formulation have any connection to generalization in the first place?

In summary, I believe this is a solid empirical work, proposing a certain heuristic approach to address certain practicality issues encountered in cross device FL models of a certain type (while clearly the approach is model-agnostic, it is not clear at all it makes sense to use it for any model type - some models might not be well trainable using alternating minimization steps).

I think this paper should be seen as borderline. I tend to accept it as the reviews are largely positive. But at the same time, I am very familiar with the related literature, and my own reading of the paper would lead to a score of 5. I will leave the final decision to SAC.

AC